# Imiquimod, a Promising Broad-Spectrum Antiviral, Prevents SARS-CoV-2 and Canine Coronavirus Multiplication Through the MAPK/ERK Signaling Pathway

**DOI:** 10.3390/v17060801

**Published:** 2025-05-31

**Authors:** Josefina Vicente, Freddy Armando Peñaranda Figueredo, Stefania Mantovani, Daniela Laura Papademetrio, Sergio Ivan Nemirovsky, Andrea Alejandra Barquero, Carina Shayo, Carlos Alberto Bueno

**Affiliations:** 1Laboratorio de Virología, Departamento de Química Biológica, Facultad de Ciencias Exactas y Naturales, Universidad de Buenos Aires, Buenos Aires 1428, Argentina; jvicente@qb.fcen.uba.ar (J.V.); ffigueredo@qb.fcen.uba.ar (F.A.P.F.); alecab@qb.fcen.uba.ar (A.A.B.); 2Instituto de Química Biológica de la Facultad de Ciencias Exactas y Naturales (IQUIBICEN), CONICET—Universidad de Buenos Aires, Buenos Aires 1428, Argentina; sin@iquibicen.fcen.uba.ar; 3Division of Clinical Immunology and Infectious Diseases, Fondazione IRCCS Policlinico San Matteo, 27100 Pavia, Italy; s.mantovani@smatteo.pv.it; 4Cátedra de Inmunología, Departamento de Microbiología, Inmunología, Biotecnología y Genética, Facultad de Farmacia y Bioquímica, Universidad de Buenos Aires, Buenos Aires 1113, Argentina; dpapademetrio@ffyb.uba.ar; 5Unidad de Conocimiento Traslacional, Hospital de Alta Complejidad del Bicentenario Esteban Echeverría, Monte Grande, Buenos Aires 1842, Argentina; 6Centro de Investigaciones en Biomedicina Traslacional CIBiMeT, CONICET—Hospital de Alta Complejidad del Bicentenario Esteban Echeverría, Monte Grande, Buenos Aires 1842, Argentina; 7Laboratorio de Patología y Farmacología Molecular, Instituto de Biología y Medicina Experimental (IBYME, CONICET), Buenos Aires 1428, Argentina; carinashayo@hotmail.com

**Keywords:** SARS-CoV-2, canine coronavirus, broad-spectrum antiviral, imiquimod, ERK, Toll-like receptors (TLRs)

## Abstract

Respiratory viruses can cause life-threatening conditions such as sepsis and acute respiratory distress syndrome. However, vaccines and effective antivirals are available for only a limited number of infections. The majority of approved antivirals are direct-acting agents, which target viral proteins essential for infection. Unfortunately, mutations have already emerged that confer resistance to these antivirals. In addition, there is an urgent need for broad-spectrum antivirals to address the unpredictable emergence of new viruses with pandemic potential. One promising strategy involves modulating the innate immune response and cellular signaling. Imiquimod, a Toll-like receptor 7 (TLR7) agonist, has shown efficacy in murine models of influenza and respiratory syncytial virus (RSV). Additionally, it demonstrates antiviral activity against herpes simplex virus type 1 (HSV-1) and RSV independent of the TLR7/nuclear factor kappa B (NF-κB) pathway, with protein kinase A (PKA) as a crucial downstream effector. In this study, we demonstrate that imiquimod exhibits concentration-dependent antiviral activity against severe acute respiratory syndrome coronavirus 2 (SARS-CoV-2) and canine coronavirus (CCoV) in epithelial cells, underscoring its broad-spectrum action against coronaviruses. Moreover, its anti-coronavirus effect appears to be independent of the TLR/NF-κB and PKA/exchange protein directly activated by cyclic adenosine monophosphate (EPAC) pathways and may instead be linked to the activation of the mitogen-activated protein kinase kinase (MEK)/extracellular signal-regulated kinase (ERK) pathway. The ability of imiquimod to inhibit coronavirus replication via the MEK/ERK pathway, coupled with its immunomodulatory properties, highlights its potential as a broad-spectrum antiviral.

## 1. Introduction

The recent outbreak of severe acute respiratory syndrome coronavirus 2 (SARS-CoV-2) has underscored the global threat that respiratory viral infections pose to public health. Respiratory viruses can cause severe, life-threatening conditions such as sepsis and acute respiratory distress syndrome [1,2]. The treatment strategies for these viruses typically focus on supportive care and antiviral medications that inhibit viral replication and dissemination in the host [3]. However, vaccines and effective antiviral drugs are available for only a limited range of viral infections. In the past 50 years, the Food and Drug Administration (FDA) has approved just 90 antiviral drugs, targeting only nine human viral diseases. In the case of SARS-CoV-2, the therapeutic options include ritonavir-boosted nirmatrelvir (Paxlovid), remdesivir, and molnupiravir. These antivirals are classified as direct-acting antiviral agents (DAAs), given that they are designed to target viral proteins essential for infection. However, mutations have already emerged that provide resistance to these antivirals, and similar challenges have been observed in other viruses [4,5,6,7,8,9]. This is partly due to the focus on targeting viral proteins that are susceptible to mutations [2]. Additionally, viruses produce a limited number of proteins, and only some of these are viable targets for drug development. In this regard, their effectiveness is frequently limited to species and, at times, even to specific strains. The coronavirus disease 2019 (COVID-19) pandemic has also highlighted the critical need for broad-spectrum antiviral drugs to address the unpredictable emergence of new viruses with pandemic potential [10,11,12].

Conversely, compounds that inhibit viral growth by targeting host cellular proteins and pathways (known as host-directed antivirals) are gaining attention [12,13,14,15]. These agents are less susceptible to viral resistance and are more likely to exhibit broad-spectrum activity, since they target host cell factors that a wide range of viruses rely on for replication. A notable strategy within this category is the modulation of the innate immune response and cellular signaling networks, which has emerged as a promising therapeutic approach for treating respiratory infections [13]. In fact, previous studies have shown that many viruses modulate host kinases and signaling pathways for their replication [16,17,18,19].

Imiquimod (IMQ), a Toll-like receptor (TLR) 7 agonist, is currently used to treat human papillomavirus (HPV) genital warts [20]. Additionally, we and others have demonstrated its effectiveness in murine models for influenza and respiratory syncytial virus (RSV) [21,22]. It exhibits antiviral activity against herpes simplex virus-1 (HSV-1) and RSV, independent of the TLR7/ nuclear factor kappa B (NF-κB) signaling pathway in vitro [22,23]. This antiviral activity against HSV-1 and RSV involves protein kinase A (PKA) as a crucial downstream effector of IMQ [22,23].

Thus, given the broad-spectrum antiviral activity and immunomodulatory effects of IMQ observed in both in vitro and in vivo studies, the current study aimed to investigate its antiviral potential against SARS-CoV-2 and canine coronavirus (CCoV). Additionally, we sought to evaluate its impact on the signaling pathways involved in its antiviral action in coronavirus-infected cells.

## 2. Materials and Methods

### 2.1. Reagents

IMQ (TLR7 ligand), Pam2CSK4 (TLR2/TLR6 ligand), Resiquimod (R848, TLR7/8 ligand), poly(I:C)-HMW (TLR3 ligand), and BAY 11-7082 were purchased from InvivoGen (San Diego, CA, USA). ESI-09 and PD98059 were acquired from Tocris Cookson Inc. (Ballwin, MO, USA). U0126 and the rabbit monoclonal antibodies anti-Phospho-CREB (Ser133) (87G3), anti-CREB (48H2), and anti-Phospho-ERK1/2 were acquired from Cell Signaling (Danvers, MA, USA). The rabbit polyclonal antibody anti-ERK1/2 was purchased from Santa Cruz Biotechnology (Dallas, TX, USA). H89, Phorbol 12-myristate 13-acetate (PMA), the rabbit polyclonal antibodies anti-Spike and anti-Nucleocapsid of SARS-CoV-2, and 4′,6-diamidino-2-phenylindole (DAPI) were obtained from Sigma-Aldrich (St. Louis, MO, USA). Secondary goat anti-rabbit FluoroLinkTMCyTM2 antibody was purchased from GE Healthcare (Chicago, IL, USA).

### 2.2. Cells and Viruses

The human Calu-3 cell line (human lung adenocarcinoma) was kindly provided by Dr. Fernanda Elias (Fundación Pablo Cassará-Buenos Aires, Argentina) and grown in Dulbecco’s modified Eagle medium (D-MEM) supplemented with 10% fetal bovine serum (FBS). Vero E6 cell line (African green monkey kidney epithelial cells) and SARS-CoV-2 Wuhan strain were kindly provided by Dr. Jorge Quarleri (INBIRS-Buenos Aires, Argentina). Vero E6 cells were grown in minimum essential medium (MEM) supplemented with 5% FBS. CCoV and CRFK cells (cat kidney cortex epithelial cells) were kindly provided by Dr. Carlos Palacios (Fundación Pablo Cassará-Buenos Aires, Argentina). CRFK cells were grown in D-MEM supplemented with 10% FBS. CCoV and SARS-CoV-2 were used and propagated at low multiplicity of infection (moi). The experiments involving SARS-CoV-2 were conducted in the Biosafety Level 3 (BSL-3) Laboratory of the operational unit for biological containment center (UOCCB) at ANLIS-Malbrán, Buenos Aires, Argentina.

### 2.3. Antiviral Activity

Virus yield inhibition assay was performed as previously described [24]. Briefly, Calu-3 and CRFK cell monolayers cultured in 24-well plates were infected with SARS-CoV-2 and CCoV, respectively, at an moi of 0.1. After 1 h adsorption at 37 °C, the inoculum was discarded, and cells were treated with the compounds for 24 h. Supernatants were subsequently collected and titrated by plaque assay in Vero E6 and CRFK cells, respectively. The effective concentration 50 (EC_50_) was defined as the concentration of the compound that reduced viral yields by 50% relative to the virus control.

### 2.4. Cytotoxicity Assay

For cell viability determination, Calu-3 and CRFK cell monolayers were cultured in 96-well plates for 24 h. Subsequently, cultures were treated with different concentrations of IMQ for 24 h. The mock treatment was considered as 100% of cell viability. All treatments were performed in triplicate. Finally, cell viability was assessed by the 3-(4,5-dimethylthiazol-2-yl)-2,5-diphenyltetrazolium bromide (MTT) (Sigma-Aldrich) assay, following the manufacturer’s instructions. Absorbance was measured at 570 nm using a BioTek ELx808 microplate reader. The concentration required to reduce cell viability by 50% (CC_50_) was calculated, as described previously [25]. The selectivity index (SI) was calculated as the ratio of CC_50_ to EC_50_ (SI = CC_50_/EC_50_).

### 2.5. Virucidal Effect

Virucidal activity assay was performed as previously described [26]. Briefly, SARS-CoV-2 and CCoV (10^6^ plaque-forming units, PFU) were incubated for 2 h at 37 °C in culture medium with or without IMQ. Samples were then diluted to a non-inhibitory drug concentration and titrated via plaque assay in Vero E6 and CRFK cells, respectively.

### 2.6. Pretreatment, Adsorption and Internalization Assays

Pretreatment, adsorption and internalization assays were performed as previously described [24]. For pre-infection assay, cells were incubated or not with IMQ (10 μg/mL) at 37 °C for 2 h, washed with PBS, and subsequently infected for 1 h at 37 °C. The inoculum was then removed; cells were supplemented with fresh medium, and further incubated for 24 h at 37 °C. To determine whether the drug affects viral adsorption, cells were infected in the presence or not of IMQ (10 μg/mL) and incubated for 1 h at 4 °C. The inoculum was then removed; cells were washed, supplemented with fresh medium, and further incubated for 24 h at 37 °C. Viral yields were collected and titrated using a plaque assay. To evaluate the impact of the drug on viral internalization, cells were infected and incubated for 1 h at 4 °C. The inoculum was discarded, and cells were washed and treated or not with IMQ (10 μg/mL) for 2 h at 37 °C. Non-internalized virus was then inactivated by exposure to citrate buffer (pH 3) for 1 min. Afterward, cells were incubated with fresh medium until 24 h post-infection (p.i.) at 37 °C. Viral yields were collected and titrated using a plaque assay.

### 2.7. Time-of-Addition and Time-of-Removal Assays

Time-of-addition and time-of-removal assays were performed as previously described with minor modifications [25,27]. Cells were infected for 1 h at 4 °C, and after removing the inoculum, cells were washed with phosphate-buffered saline (PBS) and treated with IMQ at 0, 1, 2, 4, 6, 8, 12, 18, 20 and 22 h p.i., followed by incubation at 37 °C until 24 h p.i. For the time-of-removal assay, cells were infected for 1 h at 4 °C, and after removing the inoculum, cells were washed with PBS and treated with IMQ. IMQ was removed and supplemented with fresh medium at 1, 2, 4, 6, 8, 10 and 12 h p.i., followed by incubation at 37 °C until 24 h p.i. In both assays, inhibition percentages at each time point were calculated relative to an untreated control. Virus yields were determined by plaque assay.

### 2.8. Immunofluorescence Assay (IF)

The immunofluorescence procedure was conducted as described previously [24]. Calu-3 cells were infected with SARS-CoV-2 at an moi of 0.1 and treated or not with IMQ (10 μg/mL) for 24 h. Cells were then fixed with methanol and stored at 4 °C in PBS. Expression of the Spike (S) and Nucleocapsid (N) proteins was assessed by indirect immunofluorescence assays using specific antibodies, and cell nuclei were stained with DAPI (1:1000). Observations and photographs were taken using an Olympus IX71 fluorescence microscope. Image analysis was performed using Fiji Software (version 1.53v) (National Institutes of Health, Bethesda, MD, USA). The proportion of positive cells was calculated as the ratio of viral protein-positive cells to the total number of DAPI-stained cells, based on the analysis of two coverslips and evaluating at least 6 fields.

### 2.9. Raw RNA-Seq Data Analysis

RNA-seq data used in this study were originally generated and utilized in Mantovani et al. (2021) [28]. Briefly, peripheral blood samples were collected from six individuals: three healthy donors (HDs) and three patients carrying TLR7 loss-of-function (LOF) variants. Peripheral blood mononuclear cells (PBMCs) were isolated and either stimulated with IMQ (5 µg/mL) for 4 h or left unstimulated. RNA was extracted and sequenced using the Illumina platform. Cytoscape 3.10.2 software was employed for Gene Ontology (GO) enrichment analysis to assess the functional clustering of differentially expressed genes, considering a log_2_FC > 1. Genes associated with the mitogen-activated protein kinases (MAPK) signaling pathway, identified through enrichment analysis using the Kyoto Encyclopedia of Genes and Genomes (KEGG) pathway database, were represented in a heatmap to visualize their differential expression.

### 2.10. Western Blot Analysis

The western blot analysis procedure was conducted as described previously with minor modifications [22]. Whole cell extracts from cultures were loaded onto a 10% sodium dodecyl sulfate-polyacrylamide gel electrophoresis (SDS-PAGE) gel and transferred to polyvinylidene fluoride (PVDF) membranes for 60 min at 75 mA. Membranes were blocked with PBS containing 5% milk for 1 h, followed by overnight incubation with diluted primary antibodies at 4 °C. After washing, membranes were incubated with diluted peroxidase-conjugated secondary antibodies for 1 h at room temperature with agitation. Chemiluminescence detection was performed using Biolumina (Kalium Technologies, Buenos Aires, Argentina) and immunoreactive bands were visualized with an enhanced chemiluminescence system (Amersham ECL). Cyclic adenosine monophosphate (cAMP)-response element binding protein (CREB) and extracellular signal-regulated kinase (ERK) served as loading controls. Densitometric analysis was performed with Scion Image v0.4.0.2. Protein levels were normalized to CREB or ERK levels and subsequently normalized to untreated cells, which were assigned a value of 1.

### 2.11. Transfections and Reporter Gene Assays

The transfection procedure was carried out as described previously [25]. Calu-3 and CRFK cells were transfected with an NF-κB–luciferase reporter vector or an activator protein 1 (AP-1)–luciferase reporter vector. Additionally, cells were transfected with an RSV-β–galactosidase control plasmid. After 24 h, cells were stimulated with various compounds. Transfections were performed using Lipofectamine 2000 (Invitrogen, Carlsbad, CA, USA) according to the manufacturer’s instructions. The NF-κB–luciferase reporter vector was kindly provided by Dr. Susana Silberstein (Universidad de Buenos Aires, Argentina). The AP-1–luciferase reporter vector was kindly provided by Prof. Dr. Thomas F. Schulz (Medizinische Hochschule Hannover, Germany). Reporter quantitation was performed using the Luciferase Assay System E1500 and β-Galactosidase Enzyme Assay System E2000 (Promega, Madison, WI, USA), following the manufacturer’s instructions. Measurements were obtained using a FLUOstar OPTIMA microplate reader.

### 2.12. Cytokine Determination

Calu-3 cell monolayers were stimulated with various compounds or left unstimulated for 6 h. The supernatants were collected to measure the concentrations of human cytokines interleukin-6 (IL-6) and interleukin-8 (IL-8) by enzyme-linked immunosorbent assay (ELISA), following the manufacturer’s instructions (BD OptEIATM, Becton–Dickinson, Franklin Lakes, NJ, USA). Absorbance was measured at 450 nm on a BioTek ELx808 microplate reader.

### 2.13. Statistical Analysis

Statistical analysis was performed as previously described [24]. Statistically significant differences were evaluated by one-way analysis of variance (ANOVA) with Tukey’s or Dunnett’s multiple comparison post hoc test or unpaired *t*-test, using GraphPad Prism 8.3 Software (GraphPad Software, San Diego, CA, USA), considering *p* values less than 0.05 as statistically significant.

## 3. Results

### 3.1. Antiviral Activity of IMQ Against Coronavirus In Vitro

In order to evaluate the effect of IMQ on coronavirus infections, Calu-3 and CRFK cells were infected with SARS-CoV-2 (moi = 0.1) and CCoV (moi = 0.1), respectively, and then exposed to 10 μg/mL IMQ over a period of 24 h, as previously used against RSV and HSV-1 [22,23]. As a control of the TLR7/8 signaling pathway, Resiquimod (RSQ), another TLR7/8 ligand, was evaluated at 10 μg/mL, as previously reported [22,23]. The IMQ treatment reduced the levels of infectious virus particles for both SARS-CoV-2 and CCoV, while RSQ did not show antiviral activity at the concentration tested (Figure 1A).

Next, to verify the antiviral activity of IMQ against coronaviruses, Calu-3 and CRFK cells were infected with SARS-CoV-2 (moi = 0.1) and CCoV (moi = 0.1), respectively, and were then exposed to different concentrations of IMQ for 24 h. IMQ notably diminished the infectivity of both viruses in a concentration-dependent manner (Table 1 and Figure 1B). Additionally, the CC_50_ and the selectivity index (SI), which compares CC_50_ and EC_50_ values, were determined for SARS-CoV-2 and CCoV (Table 1). Subsequently, we investigated whether its antiviral action against SARS-CoV-2 and CCoV resulted from direct inactivation of the released virus. Suspensions of SARS-CoV-2 and CCoV were exposed to various concentrations of IMQ for 2 h at 37 °C, followed by an assessment of the remaining infectivity. No reduction in viral titers was evident after the IMQ treatment compared to the untreated control, indicating a lack of virucidal activity (Figure 1C).

To corroborate the antiviral effects of IMQ, we evaluated its impact on the expression of the Spike (S) and Nucleocapsid (N) proteins during SARS-CoV-2 infection in Calu-3 cells. As shown in Appendix A, the number of fluorescent cells expressing S and N proteins was significantly reduced in the IMQ-treated samples (30% vs. 6% for the S protein and 47% vs. 12% for the N protein).

To further explore the stage of the viral cycle that was affected by IMQ, CRFK cells were treated with IMQ either before CCoV infection (pre-treatment) or during the virus entry phase (adsorption/internalization). The virus yields were then measured at 24 h p.i. IMQ did not inhibit CCoV replication when added prior to infection or during the virus entry phase (Figure 2A). Next, we conducted a time-of-addition/time-of-removal assay at various intervals post-infection. The results indicated that IMQ could inhibit the formation of infectious particles even when added at 10 h p.i. (Figure 2B). However, its inhibitory effect on viral replication diminished at later time points. Additionally, removing IMQ at or before 4 h p.i. resulted in no inhibition of viral replication (Figure 2C), whereas removing it at 6 h p.i. or later led to a significant inhibition of viral replication.

In summary, these findings demonstrate the efficacy of IMQ in inhibiting coronavirus propagation, primarily affecting the later stages of the viral cycle.

### 3.2. Anti-Coronavirus Activity of IMQ Was Independent of the NF-κB and PKA/EPAC Pathway and Dependent on the MEK/ERK Pathway

To confirm that the NF-κB pathway was not involved in the anti-coronavirus activity of IMQ, we examined NF-κB activation in Calu-3 and CRFK cells transfected with an NF-κB–luciferase reporter plasmid and treated with IMQ (TLR7 agonist), RSQ (TLR7 agonist), Pam2CSK4 (TLR 2/6 agonist), and poly(I:C)-HMW (TLR 3 agonist). Pam2CSK4 and poly(I:C)-HMW induced NF-κB activation, which was reduced by the NF-κB inhibitor BAY 11-7082 [24]; however, neither IMQ nor RSQ induced NF-κB activation (Appendix A). As expected, the antiviral activity of IMQ against CCoV and SARS-CoV-2 remained unaffected by BAY 11-7082 (Figure 3A).

Next, we investigated whether IMQ may exert its anti-coronavirus effects through the cAMP/PKA pathway, as previously reported for RSV and HSV-1 [22,23]. Given our previous findings that IMQ phosphorylates cAMP-response element binding protein (CREB), a direct substrate of PKA, in human epidermoid carcinoma cell line (HEp-2) and A549 cells [22], we initially measured the impact of IMQ on CREB phosphorylation in Calu-3 and CRFK cells. The IMQ treatment resulted in an increase in p-CREB levels within 30 min (Appendix A), suggesting that IMQ may also activate the PKA pathway in these cell lines. Then, to investigate whether IMQ exerted its antiviral activity against coronaviruses through the PKA pathway, we examined the effect of a PKA inhibitor (H89 dihydrochloride) to determine if it could mitigate the antiviral effect of IMQ. Considering that cAMP effects are also mediated by exchange protein directly activated by cAMP (EPAC) (another intracellular cAMP effector), we also evaluated the impact of an EPAC inhibitor (ESI-09) on IMQ activity. Consistent with previous studies linking the inhibition of key mediators of cAMP signaling to their antiviral effects against coronaviruses [29,30], we found that H89 and ESI-09 exhibited antiviral activity against SARS-CoV-2 and CCoV on their own (Figure 3B,C). Surprisingly, in the cells treated with these inhibitors, the antiviral activity of IMQ was not diminished. In fact, the viral titers were lower than with IMQ treatment alone; however, the difference was not statistically significant (Figure 3B,C). Thus, these findings suggest that the antiviral mechanism of IMQ against coronaviruses may not involve the cAMP/PKA/EPAC pathway, in contrast to its reported mode of action against RSV and HSV-1 [22,23].

To explore alternative signaling pathways as potential antiviral mechanisms of action for IMQ against coronaviruses, we investigated transcriptome variations in PBMCs isolated from healthy donors (HDs) and patients with previously identified loss-of-function (LOF) TLR7 variants after stimulation by IMQ [28].

As shown in Figure 4A, the Gene Ontology molecular function enrichment analysis revealed that, in HDs, pathways related to cytokine-mediated signaling, NF-κB signaling, G protein-coupled receptor activity, and kinase activity were upregulated when treated with IMQ compared to the non-stimulated control. Interestingly, the RNA-seq analysis of the patients with LOF variants showed that G protein-coupled receptor activity and kinase activity were the primary upregulated pathways. Thus, given that in both groups of patients the kinase activity was upregulated, and it has been previously reported that IMQ can activate MAPK [31], we further explored MAPK signaling using the KEGG database as a reference. We identified and confirmed that several genes associated with this pathway were upregulated in both HDs and patients with LOF stimulated with IMQ (Figure 4B).

Hence, we investigated whether inhibitors of the mitogen-activated protein kinase kinase (MEK)/ERK pathway (U0126 and PD98059) could attenuate the antiviral effect of IMQ. Indeed, we observed that in the cells treated with IMQ and both inhibitors of the MEK/ERK pathway, the antiviral effect of IMQ against SARS-CoV-2 and CCoV significantly decreased (Figure 5). In addition, an ERK activator such as Phorbol-12-myristate-13-acetate (PMA) also exerted antiviral activity against the coronaviruses (Figure 5).

In conclusion, IMQ may exert its antiviral activity against coronaviruses via MEK/ERK activation in Calu-3 and CRFK cells.

### 3.3. Role of the MEK/ERK Pathway in the Anti-Coronavirus Activity of IMQ

To further investigate the role of the MEK/ERK pathway in the antiviral activity of IMQ, we first explored whether IMQ was able to phosphorylate ERK1/2 in Calu-3 and CFRK cells at different time points. As shown in Figure 6A, IMQ induced the phosphorylation of ERK1/2 within a short period of 10 min, followed by a second wave of phosphorylation at 4–6 h. Subsequently, we evaluated whether ERK phosphorylation in cells stimulated for 6 h with IMQ was dependent on PKA activity. As shown in Figure 6B, we observed that the IMQ-induced phosphorylation of ERK1/2 was not affected by co-treatment with a PKA inhibitor (H89). As expected, the IMQ-induced ERK phosphorylation was prevented by U0126, a MEK inhibitor. Therefore, IMQ activated the MEK/ERK pathway independently of PKA.

Since ERK1/2 phosphorylation can activate the AP-1 transcription factor, we investigated whether IMQ could also activate AP-1. For that purpose, Calu-3 and CRFK cultures were transfected with an AP-1–luciferase reporter plasmid and then treated with IMQ and U0126 for 24 h. Phorbol-12-myristate-13-acetate (PMA) was used as a positive control for AP-1 activation. IMQ and PMA activated the AP-1 signaling pathway, and this activation was reduced by the MEK inhibitor (Figure 7A). Considering that the activation of ERK/AP-1 may induce cytokine production, we explored whether IMQ promoted the production of cytokines through this pathway in Calu-3 cells. IMQ stimulated the production of IL-6 and IL-8 in Calu-3 cells, and this effect was mitigated by treatment with a MEK inhibitor (Figure 7B). In contrast, as expected, treatment with an NF-κB inhibitor (BAY 11-7082) did not alter IMQ stimulation of cytokine production (Figure 7B). In conclusion, regardless of the absence of TLR7 expression in Calu-3 and CRFK cells, IMQ induced ERK phosphorylation, AP-1 transcriptional activation, and cytokine production via MEK/ERK activation in these cell lines.

Given that IMQ activated the MEK/ERK pathway in uninfected Calu-3 and CRFK cells, then we investigated this activation in the context of infection. It has been previously reported that SARS-CoV-2 transiently activates Raf/MEK/ERK signaling only in the very early infection phase (1 h p.i.) [34]. Therefore, to verify whether infection with CCoV can also activate this signaling pathway, we evaluated ERK1/2 phosphorylation at different times post-infection. Similar to what was reported for the SARS-CoV-2 infection, CCoV induced ERK1/2 phosphorylation only in the early stages of infection, up to the end of adsorption (1 h p.i.); during the rest of the viral replication cycle, phosphorylation levels remained similar to those of the uninfected controls (Figure 8).

Next, we assessed whether IMQ could alter this pattern of ERK1/2 phosphorylation in CCoV-infected cells. Interestingly, the CRFK cultures infected with CCoV (moi = 0.1) and treated with IMQ (10 μg/mL) exhibited phosphorylation of ERK1/2 within a short time of 10 min, followed by a second wave of phosphorylation at 6 h p.i., similar to the response observed when the uninfected cells were treated with IMQ (Figure 8).

Therefore, the cells infected with CCoV and SARS-CoV-2 induced ERK phosphorylation exclusively in the early stages of infection until the completion of adsorption (1 h p.i.). In contrast, the infected cells treated with IMQ showed an activation of this signaling pathway both at 10 min post-adsorption and again at 6 h p.i.

## 4. Discussion

Pandemics over the past centuries have predominantly been caused by respiratory viruses. Given the likelihood of future respiratory virus outbreaks, there is a need for broad-spectrum antiviral therapies to be prepared for the next pandemics, mainly for viruses with pandemic potential, such as coronaviruses and influenza viruses [10,11,12]. Furthermore, in the case of patients with a SARS-CoV-2 infection, even though hospitalizations and mortality have decreased since the rollout of the vaccines, it continues to be a serious problem among patients at very high risk for clinical progressions, such as immunocompromised patients or patients of advanced age. In this context, it is recommended to use antivirals for the early treatment of patients with COVID-19 who are at a high risk for disease progression [35,36,37].

Aldara 5% cream (IMQ) has been approved by the US Food and Drug Administration (FDA) for the treatment of actinic keratoses (AK), superficial basal cell carcinoma (sBCC), and HPV genital warts [20]. Recently, IMQ has also been investigated as an antiviral agent against respiratory viruses. In this context, our group, along with To et al. (2019), reported that the intranasal application of IMQ reduces peak viral replication, weight loss, and pulmonary inflammation in mice infected with both RSV and influenza virus [21,22].

In this paper, we demonstrate that IMQ exhibits concentration-dependent antiviral activity against both SARS-CoV-2 and CCoV in epithelial cells, highlighting a broad spectrum of action, as these viruses belong to different genera, Betacoronavirus and Alphacoronavirus, respectively. Although IMQ has been reported to reduce ACE2 expression in human bronchial epithelial cells [38], we did not observe an effect on viral entry. On the contrary, IMQ effectively restricts coronavirus infection after the virus enters the host cell, primarily impacting the later stages of the viral cycle. Similarly, we and Kan et al. have previously reported that IMQ suppresses RSV and HSV-1 replication when added p.i., without interfering with viral entry [22,23].

Historically, IMQ has been regarded as an antiviral molecule due to its ability to stimulate TLR7 signaling, leading to the expression of interferon-α/β and downstream interferon-stimulated genes (ISGs) [39]. However, it has been previously reported that IMQ exhibits antiviral activity against HSV-1 and RSV through the PKA pathway, independent of the TLR7/NF-κB signaling pathway [22,23]. In contrast to our prior findings that PKA inhibitors impair IMQ antiviral activity against RSV, we observed that PKA, EPAC, and NF-κB inhibitors do not affect IMQ anti-coronavirus activity. Therefore, we conclude that IMQ anti-coronavirus activity might be independent not only of the TLR/NF-κB pathway but also of the PKA/EPAC pathways. These findings challenge previous assumptions about the antiviral mechanism of IMQ and suggest a distinct mode of action for its antiviral activity against coronaviruses [22,23,39].

Interestingly, several genes associated with the MAPK signaling pathway were upregulated in both HDs and patients with LOF when stimulated with IMQ. This upregulation suggests that the MAPK pathway may play a central role in the cellular response to IMQ, regardless of the underlying differences in the TLR7 function between HDs and patients with LOF. In addition, despite the absence of TLR7 expression and activation in Calu-3 and CRFK cells [40,41], IMQ induces ERK phosphorylation, AP-1 transcriptional activation, and cytokine production via MEK/ERK activation in these cell lines. To our knowledge, this is the first report showing that MEK/ERK activation by IMQ can occur independently of TLR7. This suggests that another receptor might be involved in the MEK/ERK activation induced by IMQ. It has been previously reported that IMQ binds to adenosine receptors, a group of G protein-coupled receptors (GPCRs) [23,42]. Considering that ERK is one of the major cellular effectors activated by GPCRs [43,44], IMQ may bind to a GPCR present in these cells, thereby triggering the activation of the MEK/ERK pathway.

Regardless of the receptor involved in MEK/ERK activation by IMQ, this study shows that the antiviral efficacy of IMQ against coronaviruses may be linked to its ability to activate the MEK/ERK pathway. This conclusion is supported by the inhibition of the antiviral activity of IMQ in the presence of two different MEK/ERK inhibitors in coronavirus-infected cells. Interestingly, the antiviral activity of IMQ and its induction of ERK activation are temporally linked, given that IMQ must be present at or after 6 h p.i. to exhibit antiviral activity, and, coincidentally, a second wave of ERK phosphorylation is observed 6 h after IMQ treatment.

The Raf/MEK/ERK signaling pathway plays a key role in regulating cellular proliferation, differentiation, apoptosis, cytokine production, and immune responses [18,45]. Importantly, it has also been associated with the replication of both DNA and RNA viruses [18,32]. SARS-CoV-2 and CCoV infection induce a monophasic activation of Raf/MEK/ERK in the early phase of viral infection, with a peak of ERK phosphorylation occurring 1 h p.i., which is crucial in the initial stage of the viral life cycle. Schreiber et al. (2022) reported that inhibiting MEK/ERK signaling prior to and during the first hours of SARS-CoV-2 infection effectively reduces viral replication [34]. After this early activation, the coronavirus infection does not induce a second activation phase in the later stages of the viral cycle. Therefore, we hypothesize that although this early monophasic activation of the MEK/ERK pathway by a coronavirus infection may favor viral replication, IMQ induces a second phase of ERK activation at a later time after the infection, which could be detrimental to viral replication. In support of this, another MEK/ERK activator, PMA, also shows antiviral activity. This second phase of ERK activation induced by IMQ could lead to the modulation of the immune response and synthesis of antiviral products, such as cytokines or antiviral restriction factors, which may ultimately exert the antiviral activity. Consequently, this pathway represents a promising therapeutic and antiviral target.

Overall, the ability of IMQ to inhibit coronavirus replication via the MEK/ERK pathway, along with its immunomodulatory effects, positions IMQ as a potential therapeutic option for combating coronavirus infections. We believe further animal studies are warranted to fully harness its therapeutic potential. Importantly, previous findings showing the antiviral activity of IMQ against RSV and influenza virus, combined with the demonstration of anti-coronavirus activity in this study, highlight IMQ as a promising candidate for further development as a broad-spectrum antiviral therapy against respiratory viruses.

## Figures and Tables

**Figure 1 viruses-17-00801-f001:**
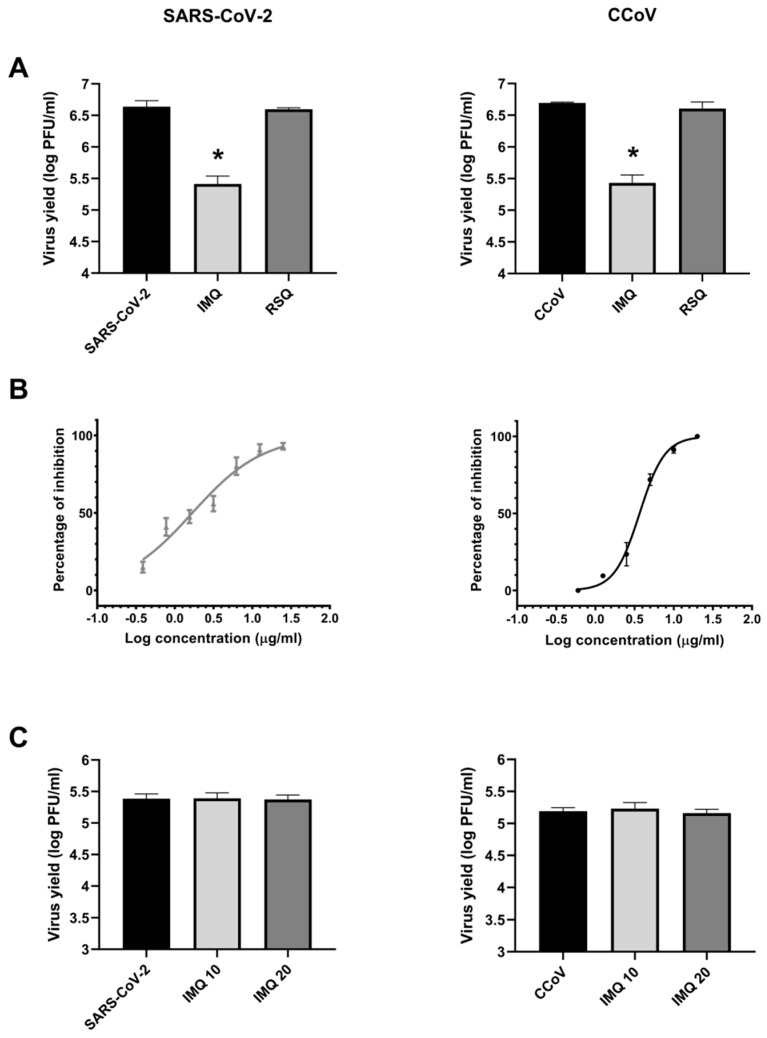
Evaluation of the anti-SARS-CoV-2 and anti-CCoV activity of IMQ in vitro. (**A**) Calu-3 and CRFK cells were infected with SARS-CoV-2 and CCoV (moi = 0.1) for 1 h, respectively. After infection, cells were treated or not with IMQ (10 μg/mL) and RSQ (10 μg/mL) for 24 h. Total virus yields were determined by plaque assay in Vero E6 and CRFK cells. (**B**) Calu-3 and CRFK cells were infected with SARS-CoV-2 and CCoV (moi = 0.1) for 1 h, respectively. After infection, cells were treated with different concentrations of IMQ. After 24 h, total virus yields were titrated by plaque assay in Vero E6 and CRFK cells. (**C**) SARS-CoV-2 and CCoV (10^6^ PFU) were incubated or not with IMQ (10 and 20 μg/mL) for 2 h at 37 °C. Remaining infectivity was determined by plaque assay in Vero E6 and CRFK cells. Data represent mean ± SD for *n* = 3 independent experiments, performed in duplicate. * Significantly different from untreated–infected control cells (SARS-CoV-2 or CCoV) (*p*-value < 0.05); one-way ANOVA with Tukey’s post test.

**Figure 2 viruses-17-00801-f002:**
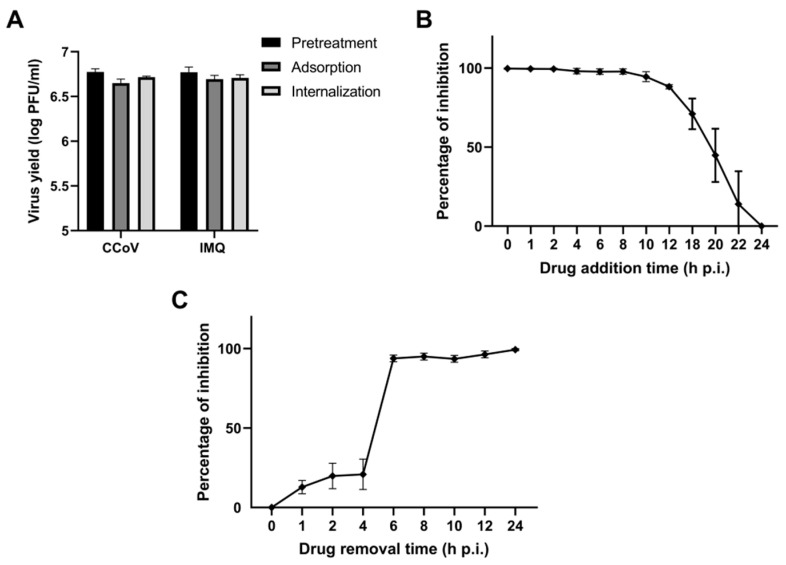
Influence of the duration of IMQ treatment on CCoV infectivity. (**A**) Pre-treatment assay: CRFK cells were treated or not with IMQ (10 μg/mL) for 2 h at 37 °C and then infected with CCoV (moi = 0.1) for 1 h at 37 °C. The inoculum was removed and replaced with fresh medium and incubated for 24 h. Virus adsorption: CRFK cells were infected with CCoV (moi = 0.1) with or without IMQ (10 μg/mL) and incubated for 1 h at 4 °C. The inoculum was discarded, and cells were supplemented with fresh medium, and further incubated for 24 h at 37 °C. Virus internalization: CRFK cells were infected with CCoV (moi = 0.1) and incubated for 1 h at 4 °C. Cells were then treated or not with IMQ (10 μg/mL) for 2 h at 37 °C. Uninternalized virus was inactivated using citrate buffer (pH 3) for 1 min. Cells were then incubated with fresh medium for 24 h p.i. at 37 °C. Virus yields were collected and titrated by plaque assay. (**B**) CRFK cells infected with CCoV (moi = 0.1) were treated or not with IMQ (10 μg/mL) at 0, 1, 2, 4, 6, 8, 12, 18, 20, and 22 h p.i. Total virus yields were determined by plaque assay in CRFK cells at 24 h p.i and plotted as the percentage of inhibition with respect to untreated–infected control. (**C**) CRFK cells were infected with CCoV (moi = 0.1) for 1 h at 4 °C. After removing the inoculum, cells were treated or not with IMQ (10 μg/mL). Then, medium with IMQ was removed and supplemented with fresh medium at 1, 2, 4, 6, 8, 10, and 12 h p.i., followed by incubation at 37 °C until 24 h p.i. The supernatants were titrated in CRFK cells using the plaque formation method. Inhibition percentages at each time point were calculated relative to untreated–infected control. Data represent mean ± SD for *n* = 3 independent experiments, performed in duplicate.

**Figure 3 viruses-17-00801-f003:**
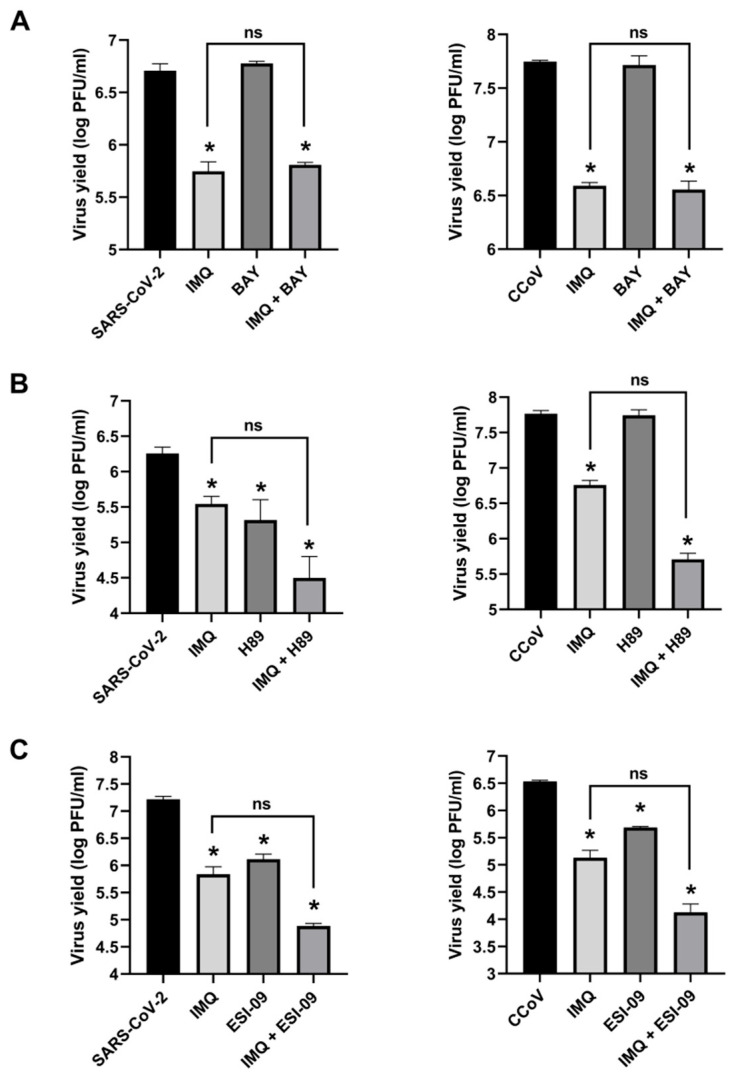
Role of NF-κB and cAMP pathways in antiviral activity of IMQ. Calu-3 and CRFK cells infected with SARS-CoV-2 (moi = 0.1) and CCoV (moi = 0.1), respectively, were treated or not with IMQ (10 μg/mL) and with (**A**) the NF-κB inhibitor BAY 11-7082 (BAY 10 μg/mL), (**B**) the PKA inhibitor H89 (10 μM), and (**C**) and the EPAC inhibitor ESI-09 (5 μg/mL) for 24 h. Supernatants were titrated by plaque assay. Data represent mean ± SD for *n* = 3 independent experiments, performed in duplicate. * Significantly different from untreated–infected control cells (SARS-CoV-2 or CCoV) (*p*-value < 0.05); ns: not significantly different from IMQ-treated cells (*p*-value ≥ 0.05); one-way ANOVA with Tukey’s post test.

**Figure 4 viruses-17-00801-f004:**
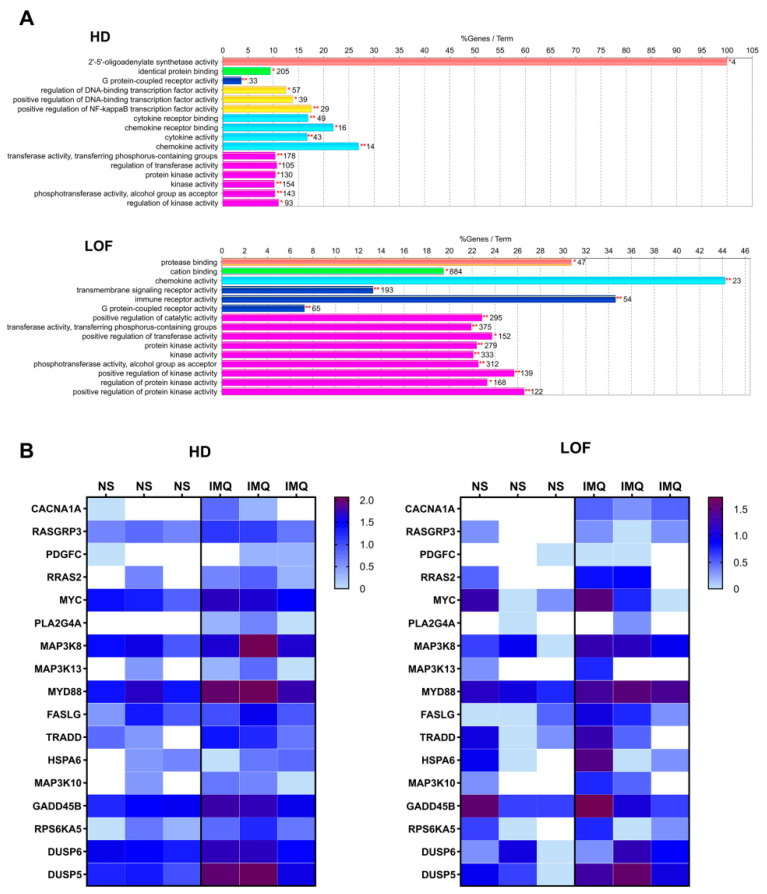
Transcriptomic analysis of molecular functions and gene expression in the MAPK pathway. (**A**) Representation of the Gene Ontology (GO) molecular function enrichment analysis performed in Cytoscape using overexpressed genes (log_2_FC > 1). Enriched terms are indicated with their representative percentage and the number of associated genes. The bars represent the percentage of overexpressed genes relative to the total number of genes associated with each molecular function in the GO database. The numbers next to the bars indicate the total number of genes linked to each molecular function in the analyzed dataset. Colors represent GO terms grouped by functional similarity, with each color corresponding to a distinct functional group. * denote statistically significant enrichment after multiple testing correction (adjusted *p*-value < 0.05) ** denote statistically significant enrichment after multiple testing correction (adjusted *p*-value < 0.01). (**B**) Heatmap of log-transformed counts per million (logCPM) for genes belonging to the MAPK signaling pathway identified by KEGG in patients with TLR7 loss-of-function (LOF) and healthy donors (HDs) for both non-stimulated (NS) and imiquimod-stimulated (IMQ) conditions.

**Figure 5 viruses-17-00801-f005:**
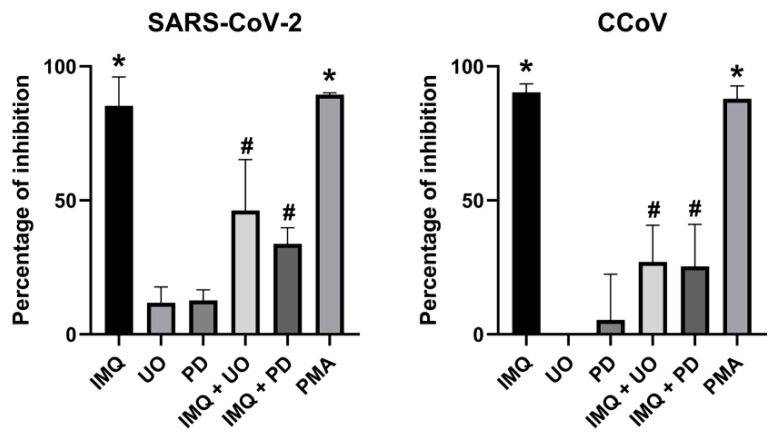
Impact of MEK/ERK activity on the antiviral effects of IMQ. Calu-3 and CRFK cells infected with SARS-CoV-2 (moi = 0.1) and CCoV (moi = 0.1), respectively, were treated or not with IMQ (10 μg/mL), (U0, 15 μM), PD98059 (PD, 25 μM), and PMA (200 nM) for 24 h at 37 °C, using concentrations previously reported to be effective [32,33]. Supernatants were titrated by plaque assay and plotted as the percentage of inhibition with respect to untreated–infected control. Data represent mean ± SD for *n* = 3 independent experiments, performed in duplicate. * Significantly different from untreated–infected control (*p*-value < 0.05); # significantly different from IMQ treated cells (*p*-value < 0.05); one-way ANOVA with Tukey’s post test.

**Figure 6 viruses-17-00801-f006:**
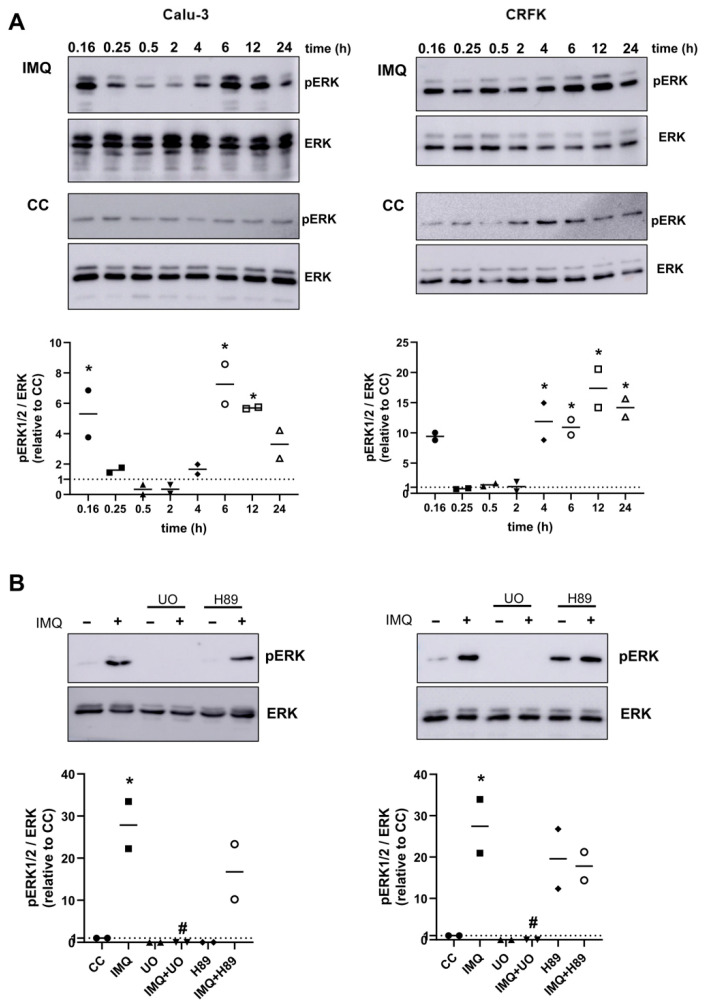
Effect of IMQ on ERK phosphorylation. (**A**) Immunoblot images of p-ERK and ERK expression and quantitative densitometric analysis in Calu-3 and CRFK cells after exposure or not to IMQ (10 μg/mL) during the indicated times. (**B**) Immunoblot images of p-ERK and ERK expression and quantitative densitometric analysis in Calu-3 and CRFK cells after exposure or not to IMQ (10 μg/mL), U0126 (U0, 15 μM) and H89 (10 μM) for 6 h. (CC): unstimulated control cells. Solid line represents mean for *n* = 2 independent experiments. * Significantly different from CC (*p*-value < 0.05); # significantly different from IMQ treated cells (*p*-value < 0.05); one-way ANOVA with Tukey’s post test.

**Figure 7 viruses-17-00801-f007:**
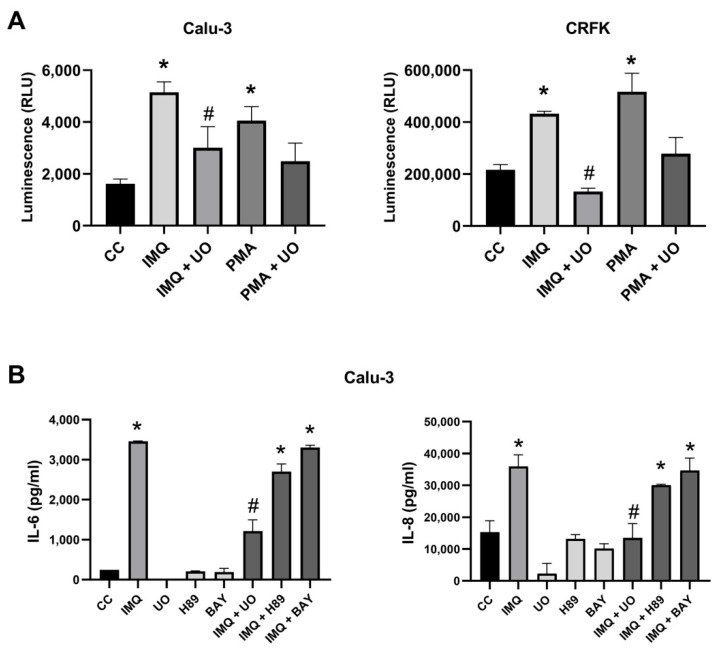
Effects of IMQ on AP-1 activation and cytokine production. (**A**) Calu-3 and CRFK cells were transfected with AP-1–luciferase reporter plasmid and β-galactosidase control plasmid. After 24 h, cells were treated or not with IMQ (10 μg/mL), PMA (200 nM), and U0126 (U0, 15 μM) for 24 h. Luciferase activity was measured in cell extracts, and each value was normalized to β-galactosidase activity in relative luciferase units (RLUs). (**B**) Calu-3 cells were treated or not with IMQ (10 μg/mL), U0126 (U0, 15 μM), H89 (BAY, 10 μM), and BAY 11-7082 (10 μg/mL) for 6 h. IL-6 and IL-8 were determined by ELISA. CC: unstimulated control cells. Data represent mean ± SD for *n* = 3 independent experiments, performed in triplicate. * Significantly different from CC (*p*-value < 0.05); # significantly different from IMQ treated cells (*p*-value < 0.05); one-way ANOVA with Tukey’s post test.

**Figure 8 viruses-17-00801-f008:**
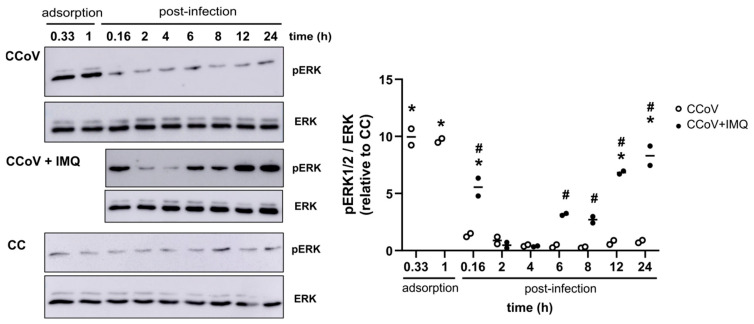
Effect of CCoV infection and IMQ on ERK phosphorylation. Immunoblot images of p-ERK and ERK expression and quantitative densitometric analysis in CRFK cells after infection with CCoV (moi = 0.1) and exposure or not to IMQ (10 μg/mL) during the indicated times. CC: unstimulated control cells. Solid line represents for *n* = 2 independent experiments. * Significantly different from CC (*p*-value < 0.05); # significantly different from CCoV (*p*-value < 0.05); one-way ANOVA with Tukey’s post test.

**Table 1 viruses-17-00801-t001:** EC_50_ and SI of IMQ against SARS-CoV-2 and CCoV.

	SARS-CoV-2	CCoV
Cell Line	Calu-3	CRFK
EC_50_ (μg/mL)	1.69 ± 1.11	3.66 ± 1.03
CC_50_ (μg/mL)	372.6 ± 1.06	95.97 ± 1.10
SI (CC_50_/CE_50_)	219.43	26.18

Effective Concentration 50 (EC_50_, μg/mL) and Cytotoxic Concentration 50 (CC_50_, μg/mL) were calculated by nonlinear regression. SI: selectivity index (ratio CC_50_/CE_50_). Data represent mean ± SD for *n* = 3 independent experiments, performed in triplicate.

## Data Availability

The data presented in this study are available on request from the corresponding author.

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
