# Peer review of "Imiquimod, a Promising Broad-Spectrum Antiviral, Prevents SARS-CoV-2 and Canine Coronavirus Multiplication Through the MAPK/ERK Signaling Pathway"

_viruses, 2025, doi:10.3390/v17060801_

Round 1

Reviewer 1 Report

Comments and Suggestions for Authors

The present report by Vicente et al aims at simultaneously introduce the potential use of Imiquimod as an antiviral agent for SARS-CoV-2 while presenting a preliminary description of the potential underlying mechanisms.

The potential use of Imiquimod for SARS-CoV-2 has been reported previously, both alone or in combination with vaccines (see for example https://doi.org/10.1038/s41392-023-01485-6, https://www.frontiersin.org/journals/immunology/articles/10.3389/fimmu.2021.743890/full and papers by Serafín-Lopez).

This report builds on previous results obtained by Bueno's group, demonstrating effectiveness of imiquimod in vitro. The paper is well structured, and the results and data are sound.

A few remarks:

  1. Considering that the effect of imiquimod on ACE2 expression levels has been previously reported, have you considered analyzing if at least part of the observed effect can be explained by this pathway? If you don't have the means to conduct an experiment to delve into this, at least this potential effect should be mentioned and discussed
  2. On Figure 1, panel B: The fit on the right curve is not good. Not a single point is remotely closed to the sigmoideal fit. Could the authors try to improve on this? Or at least comment on what might happening for such marked deviations to take place
  3. The meaning of EC50 and CC50 should be explained in the text for a wide audience.

Author Response

1) Considering that the effect of imiquimod on ACE2 expression levels has been previously reported, have you considered analyzing if at least part of the observed effect can be explained by this pathway? If you don't have the means to conduct an experiment to delve into this, at least this potential effect should be mentioned and discussed.

Response: We thank the reviewer for pointing this out. Initially, we had not considered whether the antiviral effect could be explained by this pathway, as that would suggest that IMQ affects viral entry. However, our observations indicate that IMQ effectively restricts coronavirus infection after the virus has entered the host cell, primarily impacting the later stages of the viral cycle. Therefore, as suggested by the reviewer, we have now addressed and discussed this point in the Discussion section of the revised manuscript, which reads as follows: “Although IMQ has been reported to reduce ACE2 expression in human bronchial epithelial cells [38], we did not observe an effect on viral entry. On the contrary, IMQ effectively restricts coronavirus infection after the virus enters the host cell, primarily impacting the later stages of the viral cycle.”

2) On Figure 1, panel B: The fit on the right curve is not good. Not a single point is remotely closed to the sigmoideal fit. Could the authors try to improve on this? Or at least comment on what might happening for such marked deviations to take place

Response: Thank you for pointing out the mismatch between the data points and the sigmoidal fit in Figure 1, panel B. Upon closer inspection, we realized that the initial nonlinear regression equation did not adequately capture the experimental response. We have now updated the fit using a more appropriate nonlinear regression model with a variable slope, which better reflects the data. This adjustment resulted in a slight change in the calculated EC50 value (3.41 µg/ml vs. 3.66 µg/ml), which was corrected in the corresponding table.

3) The meaning of EC50 and CC50 should be explained in the text for a wide audience.

Response: We agree with this comment and have accordingly added an explanation of the meaning of EC50 in the Materials and Methods section of the revised paper (Section 2.3). The definition of CC50 was already present in the Materials and Methods section (Section 2.4). We have also included a definition of the selectivity index (SI).

Reviewer 2 Report

Comments and Suggestions for Authors

The authors describe antiviral activity of Imiquimod against two CoVs, overall the manuscript is sound and interesting.

Comments:

  • The title states that IMQ exhibits antiviral activity against CoVs but only two were tested. These represent alpha and beta CoVs but I believe more viruses would need to be tested to support the CoV activity statement. Recommend specifying SARS-CoV-2 and CCoV in the title.
  • L25, it is stated that existing antivirals only target viral proteins. How do the authors reconcile this statement with this review (for example): PMID 32404434.
  • L44, I think it is more accurate to state that there has been one outbreak of SARS-CoV-2; i.e. one spillover.
  • CCoV is only defined in Methods, I suspect many readers may not be familiar with this virus and it should be defined earlier in abstract or intro.
  • L54, provide references for resistance mutations.
  • L69, refs 9, 10 only refer to ERK/MAPK but the statement is broader. Either refine statement or ref a broader review.
  • L71-74, this sentence might be easier to read if split.
  • The literature referenced shows in vivo antiviral activity of IMQ in mice, and in vitro activity against RSV. My reading of the RSV paper is that only pre-treatment of IMQ was tested, so is this the first report of in vitro activity DURING virus replication? Please clarify this.
  • L215, please provide references for how concs of IMQ and RSQ were used.
  • Fig 1: what does CV stand for? Control virus? I think the figures would be easier to read if CV was just replaced with SARS-CoV-2 and CCoV.
  • Fig 1: I recommend presenting the data to show where the individual replicates lie, for example a jitter plot. Especially as some experiments only have an n=2, so it's important to see the spread of the data points.
  • L293, have you done any experiment to show that BAY 11-7082 is active?
  • Luminescence is misspelt throughout the manuscript, unless I'm unaware of an alternative spelling.
  • SupF3, I don't find CC a very intuitive abbreviation, esp CC50 is used elsewhere. Also, the 30m claim for Calu-3 cells is not at all convincing. Either remove claim for Calu-3 (CRFK is fine) or show more gels/replicates.
  • Fig 3, the authors claim (L309) that BAY does not diminish the antiviral effect of IMQ but not statistical test has been performed between these two conditions in the figure, as the legend states "*significantly different from CV". 
  • L358, please provide references for use of drugs at specified concentrations.
  • L386, UO126 should be U0126.
  • L391, authors state "IMQ stimulated the production of IL-6 and IL-8 in Calu-3 cells in a concentration-dependent manner" but only one conc of IMQ was used.
  • L4017, not infected=uninfected.
  • F8, I'm not clear on how ERKp was normalised. Normalised both to ERK AND relative to CC? Please clarify this, maybe a formula in the methods.
  •  

Author Response

1) The title states that IMQ exhibits antiviral activity against CoVs but only two were tested. These represent alpha and beta CoVs but I believe more viruses would need to be tested to support the CoV activity statement. Recommend specifying SARS-CoV-2 and CCoV in the title.

Response: We agree with this comment and have accordingly specified both SARS-CoV-2 and canine coronavirus in the title.

2) L25, it is stated that existing antivirals only target viral proteins. How do the authors reconcile this statement with this review (for example): PMID 32404434.

Response: We thank the reviewer for pointing this out. In the cited review (PMID: 32404434), the authors state that “The majority of the antiviral drugs that have been approved by the Food and Drug Administration (FDA) act by directly targeting virus-encoded factors”. We have clarified this point and revised the corresponding sentence in the abstract. It now reads: “The majority of approved antivirals are direct-acting agents, which target viral proteins essential for infection”. Additionally, we have included this reference (PMID: 32404434) in the Introduction section of the revised manuscript.

3) L44, I think it is more accurate to state that there has been one outbreak of SARS-CoV-2; i.e. one spillover.

Response: We agree with this comment and have now revised the sentence to the singular form: “The recent outbreak of SARS-CoV-2 has underscored the global threat that respiratory viral infections pose to public health.”

4) CCoV is only defined in Methods, I suspect many readers may not be familiar with this virus and it should be defined earlier in abstract or intro.

Response: As requested by the reviewer, we have now defined CCoV in the introduction.

5) L54, provide references for resistance mutations.

Response: As requested by the reviewer, we have now provided references related to resistance mutations.

6) L69, refs 9, 10 only refer to ERK/MAPK but the statement is broader. Either refine statement or ref a broader review.

Response: As requested by the reviewer, we have now provided additional references that include a broader range of signaling pathways and viruses.

7) L71-74, this sentence might be easier to read if split

Response: We agree with the reviewer, and we have now split this sentence.

8) The literature referenced shows in vivo antiviral activity of IMQ in mice, and in vitro activity against RSV. My reading of the RSV paper is that only pre-treatment of IMQ was tested, so is this the first report of in vitro activity DURING virus replication? Please clarify this.

Response: In our study on RSV, we demonstrated the antiviral activity of IMQ against RSV when it was added after infection. In contrast, pre-treatment with IMQ or treatment during viral adsorption did not exhibit antiviral effects. These findings align with the results obtained in the present study against SARS-CoV-2 and CCoV, where we also observed antiviral activity when IMQ was added post-infection, and IMQ did not inhibit CCoV replication when administered prior to infection or during the viral entry phase. We have now clarified this point in the Discussion section of the revised manuscript, which reads as follows: “IMQ effectively restricts coronavirus infection after the virus enters the host cell, primarily impacting the later stages of the viral cycle. Similarly, we and Kan et al. have previously reported that IMQ suppresses RSV and HSV-1 replication when added post-infection, without interfering with viral entry.”

9) ”L215, please provide references for how concs of IMQ and RSQ were used.

Response: As requested by the reviewer, we have now provided references for the use of IMQ and RSQ at the specified concentrations. The revised text reads as follows: “In order to evaluate the effect of IMQ on coronaviruses infection, Calu-3 and CRFK cells were infected with SARS-CoV-2 (moi = 0.1) and CCoV (moi = 0.1), respectively, and then exposed to IMQ 10 μg/ml during 24 h, as previously used against RSV and HSV-1 [22,23]. As a control of the TLR7/8 signaling pathway, resiquimod (RSQ), another TLR7/8 ligand, was evaluated at 10 μg/ml, as previously reported [22,23]

10) Fig 1: what does CV stand for? Control virus? I think the figures would be easier to read if CV was just replaced with SARS-CoV-2 and CCoV.

Response: As requested by the reviewer, we have now replaced "CV (Control Virus)" with "SARS-CoV-2" and "CCoV" in all the figures.

11) Fig 1: I recommend presenting the data to show where the individual replicates lie, for example a jitter plot. Especially as some experiments only have an n=2, so it's important to see the spread of the data points.

Response: As requested by the reviewer, we have now presented the data from experiments with n=2 as jitter plots. This includes the western blots shown in Figure 6, Figure 8, and Supplementary Figure 3.

12) L293, have you done any experiment to show that BAY 11-7082 is active?

 Response: BAY 11-7082 is a well-known inhibitor of the NF-κB pathway. We have been using BAY 11-7082 (InvivoGen) in our laboratory for a long time and have conducted numerous experiments demonstrating its activity in our experimental settings. In fact, we have previously published results using this compound as a positive control to validate cytokine modulation and NF-κB activation (Vicente et al, Viruses 2023, 15(4), 989). We thank the reviewer for pointing out that we did not include experimental evidence of BAY 11-7082 activity in the current manuscript. To address this, we have now included data demonstrating its inhibitory effect on NF-κB activation induced by Pam2CSK4 and poly(I:C)-HMW in Calu-3 and CFRK cells (Supplementary Figure 2). Additionally, we have cited a previously published study where we observed its activity (Vicente et al, Viruses 2023, 15(4), 989). Accordingly, the Results section of the revised manuscript now reads: “Pam2CSK4 and poly(I:C)-HMW induced NF-κB activation, which was reduced by the NF-κB inhibitor BAY 11-7082 [24]; however, neither IMQ nor RSQ induced NF-κB activation (Supplementary Figure 2).”

 13) Luminescence is misspelt throughout the manuscript, unless I'm unaware of an alternative spelling. 

Response: We apologize for the mistake, and it has now been corrected.

14) SupF3, I don't find CC a very intuitive abbreviation, esp CC50 is used elsewhere. Also, the 30m claim for Calu-3 cells is not at all convincing. Either remove claim for Calu-3 (CRFK is fine) or show more gels/replicates.

Response: We agree with the reviewer and have now replaced "CC" with "time = 0" in Supplementary Figure 3. Regarding the 30-minute claim for Calu-3 cells, we believe the statistical analysis is robust, and we have previously demonstrated this effect also in A549 and Hep-2 cells (Salinas et al, Antiviral Res. 2020:179:104817). Additionally, we have obtained similar results in other cell lines, such as Vero and J774A.1 cells (unpublished data). Thus, this appears to be a common pattern induced by IMQ, irrespective of the cell line. Nevertheless, we have now included another replicate that more clearly demonstrates this effect in Calu-3 cells.

15) Fig 3, the authors claim (L309) that BAY does not diminish the antiviral effect of IMQ but not statistical test has been performed between these two conditions in the figure, as the legend states "*significantly different from CV".

Response: We thank the reviewer for pointing this out. We had already performed the statistical test comparing “IMQ-treated cells” and “IMQ + BAY-treated cells” and found no statistically significant differences between them. To make this clear, we have now included the initials ns in Figure 3A between these two groups, indicating “ns: not significantly different from IMQ-treated cells.” We have also added these initials in Figures 3B and 3C to show that there were no significant differences between “IMQ-treated cells” and cells treated with “IMQ + H89” or “IMQ + ESI-09,” respectively

16) L358, please provide references for use of drugs at specified concentrations.

Response: As requested by the reviewer, we have now provided references for use of U0126, PMA and PD98059 at specified concentrations.

17) L386, UO126 should be U0126.

Response: We apologize for the mistake, and it has now been corrected.

18) L391, authors state "IMQ stimulated the production of IL-6 and IL-8 in Calu-3 cells in a concentration-dependent manner" but only one conc of IMQ was used.

Response: We thank the reviewer for pointing this out. The phrase “concentration-dependent manner” has now been removed, as only one concentration of IMQ was used.

19) L407, not infected=uninfected.

Response: We thank the reviewer for pointing this out, and it has now been replaced.

20) F8, I'm not clear on how ERKp was normalised. Normalised both to ERK AND relative to CC? Please clarify this, maybe a formula in the methods.

Response: We thank the reviewer for pointing this out. We have now clarified how ERKp was normalized. Each ERKp value was first normalized to ERK levels, and then further normalized to the control condition (CC), which was arbitrarily assigned a value of 1. Accordingly, the revised Materials and Methods section now reads: “Densitometric analysis was performed with Scion Image v0.4.0.2. Protein levels were normalized to CREB or ERK levels and subsequently normalized to untreated cells, which wer
